Can macroalgae provide promising anti-tumoral compounds? A closer look at Cystoseira tamariscifolia as a source for antioxidant and anti-hepatocarcinoma compounds

Vizetto-Duarte Catarina 1
Custódio Luísa 1
Acosta Gerardo 2 3
Lago João H.G. 4
Morais Thiago R. 4
Bruno de Sousa Carolina 1
Gangadhar Katkam N. 1 5
Rodrigues Maria João 1
Pereira Hugo 1
Lima Raquel T. 6 7 8
Vasconcelos M. Helena 6 7 9
Barreira Luísa 1
Rauter Amélia P. 10
Albericio Fernando 2 3 11
Varela João 1 jvarela@ualg.pt
1 Centre of Marine Sciences, Faculty of Sciences and Technology, Campus of Gambelas, University of Algarve , Faro , Portugal
2 Institute for Research in Biomedicine of Barcelona, Chemistry and Molecular Pharmacology , Barcelona Science Park, Baldiri Reixac, Barcelona , Spain
3 CIBER-BNN, Networking Centre on Bioengineering, Biomaterials and Nanomedicine , Barcelona Science Park, Baldiri Reixac, Barcelona , Spain
4 Institute of Environmental, Chemical and Pharmaceutical Sciences, Federal University of Sao Paulo , São Paulo , Brazil
5 Instituto de Tecnologia Química e Biológica, Universidade Nova de Lisboa , Lisbon , Portugal
6 i3S—Instituto de Investigação e Inovação em Saúde da Universidade do Porto , Rua Alfredo Allen, Porto , Portugal
7 IPATIMUP—Institute of Molecular Pathology and Immunology of the University of Porto, Cancer Drug Resistance Group , Rua Júlio Amaral de Carvalho, Porto , Portugal
8 Faculty of Medicine of the University of Porto, Department of Pathology and Oncology, Alameda Prof. Hernâni Monteiro , Porto , Portugal
9 Department of Biological Sciences, Faculty of Pharmacy, University of Porto, Rua de Jorge Viterbo Ferreira Porto , Portugal
10 Center of Chemistry and Biochemistry, Department of Chemistry and Biochemistry, Faculty of Sciences, University of Lisbon , Lisbon , Portugal
11 Department of Organic Chemistry, University of Barcelona , Martí i Franqués, Barcelona , Spain
Esteban María Ángeles
Electronic publication date: 2016 Feb 16
Publication date: 2016
Volume: 4
Electronic Location ID: e1704
Received 2015 Nov 23; Accepted 2016 Jan 27
Copyright: ©2016 Vizetto-Duarte et al.
Copyright year: 2016
Copyright holder: Vizetto-Duarte et al.
License: This is an open access article distributed under the terms of the Creative Commons Attribution License, which permits unrestricted use, distribution, reproduction and adaptation in any medium and for any purpose provided that it is properly attributed. For attribution, the original author(s), title, publication source (PeerJ) and either DOI or URL of the article must be cited.
License URL: https://creativecommons.org/licenses/by/4.0/

Keywords: Marine natural products, Brown algae, Antioxidant, Anti-hepatocarcinoma, Cystoseira

Funding: SEABIOMED PTDC/MAR/103957/2008 XtremeBio PTDC/MAR-EST/4346/2012 Foundation for Science and Technology and Portuguese National Budget CCMAR/Multi/04326/2013 FCT Doctoral Research Fellow SFRH/BD/81425/2011 SFRH/BD/78062/2011 SFRH/BD/105541/2014 SFRH/BPD/81882/2011 SFRH/BPD/68787/2010 FCT Investigator Programme IF/00049/2012 Financial support for this work was provided by the SEABIOMED (PTDC/MAR/103957/ 2008), XtremeBio (PTDC/MAR-EST/4346/2012) and CCMAR/Multi/04326/2013 funded by the Foundation for Science and Technology (FCT) and the Portuguese National Budget. IPATIMUP integrates the i3S Research Unit, which is partially supported by FCT. Vizetto-Duarte C. is a FCT doctoral research fellow (SFRH/BD/81425/2011) as well as Bruno de Sousa C. (SFRH/BD/78062/2011) and Pereira H. (SFRH/BD/105541/2014). Katkam N. Gangadhar (SFRH/BPD/81882/2011) and Raquel T. Lima (SFRH/BPD/68787/ 2010) are post-doctoral research fellows. Custódio L. was supported by FCT Investigator Programme (IF/00049/2012). Morais T.R. and Lago J.H.G. acknowledge CAPES and CNPq, respectively, for scientific research support. The funders had no role in study design, data collection and analysis, decision to publish, or preparation of the manuscript.

==============================
Marine organisms are a prolific source of drug leads in a variety of therapeutic areas. In the last few years, biomedical, pharmaceutical and nutraceutical industries have shown growing interest in novel compounds from marine organisms, including macroalgae. Cystoseira is a genus of Phaeophyceae (Fucales) macroalgae known to contain bioactive compounds. Organic extracts (hexane, diethyl ether, ethyl acetate and methanol extracts) from three Cystoseira species (C. humilis, C. tamariscifolia and C. usneoides) were evaluated for their total phenolic content, radical scavenging activity against 2,2-diphenyl-1-picrylhydrazyl (DPPH) and 2,2′-azino-bis(3-ethylbenzothiazoline-6-sulphonic acid) (ABTS) radicals, and antiproliferative activity against a human hepatocarcinoma cell line (HepG2 cells). C. tamariscifolia had the highest TPC and RSA. The hexane extract of C. tamariscifolia (CTH) had the highest cytotoxic activity (IC50 = 2.31 µg/mL), and was further tested in four human tumor (cervical adenocarcinoma HeLa; gastric adenocarcinoma AGS; colorectal adenocarcinoma HCT-15; neuroblastoma SH-SY5Y), and two non-tumor (murine bone marrow stroma S17 and human umbilical vein endothelial HUVEC) cell lines in order to determine its selectivity. CTH strongly reduced viability of all tumor cell lines, especially of HepG2 cells. Cytotoxicity was particularly selective for the latter cells with a selectivity index = 12.6 as compared to non-tumor cells. Incubation with CTH led to a 2-fold decrease of HepG2 cell proliferation as shown by the bromodeoxyuridine (BrdU) incorporation assay. CTH-treated HepG2 cells presented also pro-apoptotic features, such as increased Annexin V/propidium iodide (PI) binding and dose-dependent morphological alterations in DAPI-stained cells. Moreover, it had a noticeable disaggregating effect on 3D multicellular tumor spheroids. Demethoxy cystoketal chromane, a derivative of the meroditerpenoid cystoketal, was identified as the active compound in CTH and was shown to display selective in vitro cytotoxicity towards HepG2 cells.

Introduction

Macroalgae are used as food and feed, and also as sources of bioactive metabolites. In particular, Phaeophyceae algae have high contents of polysaccharides, minerals, polyunsaturated fatty acids and vitamins (Balboa et al., 2013). Furthermore, these organisms contain high levels of secondary metabolites with pharmacological interest, such as terpenoids, phenolic compounds and alkaloids, which have been linked to interesting biomedical activities, including antitumoral and neuroprotective (Smit, 2004; Blunt et al., 2014). Among Phaeophyceae, the Cystoseira genus comprises a large number of species widely distributed in the Atlantic and Mediterranean Sea (Guiry & Guiry, 2015). Phytochemical studies have revealed that species belonging to this genus are rich in phlorotannins, sterols, meroditerpenoids and sesquiterpenoids (Amico, 1995; Moreno et al., 1998; Khanavi et al., 2012; Montero et al., 2014), some of which exhibiting antioxidant, antitumoral, antifouling and/or antimicrobial activities with potential applications in the pharmaceutical industry (Amico, 1995; Gouveia et al., 2013; Valls & Piovetti, 1995).

Phaeophyceae algae have already shown interesting biomedical properties such as Dictyota ciliolata, Padina sanctae-crucis, Turbinaria tricostata and Petalonia fascia with antiproliferative activity in cancer cell lines (Caamal-Fuentes et al., 2014; Kurt et al., 2014). Cystoseira and Fucus genus are also known to contain molecules with antioxidant properties (Mhadhebi et al., 2011; Heffernan et al., 2015; Hadj Ammar et al., 2015). Bearing in mind the high biotechnological potential of brown algae, in this work we evaluated the total phenolic contents and antioxidant activity of organic extracts of C. tamariscifolia, C. humilis and C. usneoides. The anti-proliferative potential was screened on human hepatocellular carcinoma HepG2 cells, a cell line known to be recalcitrant to cytotoxic drugs (Liu et al., 2010). The most bioactive extract (C. tamariscifolia hexane extract; named CTH) was also evaluated in several other human tumor cell lines and compared to non-tumor cells used as selectivity controls. Cytotoxicity was then further studied in terms of its action on cell proliferation inhibition and apoptosis induction, important features for potential anti-cancer therapies. It was also evaluated its effect on multicellular tumor spheroids (MCTS). This extract was then subjected to a bioactivity-guided fractionation to afford the meroditerpene demethoxy cystoketal chromane, which was bioactive against HepG2 cell line.

Material and Methods

General

Hexane, ethyl acetate (EtOAc) and diethyl ether were from Prolabo (VWR International, Leuven, Belgium). Roswell Park Memorial Institute medium (RPMI), Dulbecco’s Modified Eagle’s medium (DMEM), fetal bovine serum (FBS), L-glutamine and penicillin/streptomycin were obtained from Lonza Ibérica (Barcelona, Spain). 2,2′-azino-bis(3-ethylbenzothiazoline-6-sulphonic acid (ABTS) and 3-(4,5-dimethylthiazol-2-yl)-2,5-diphenyltetrazolium bromide (MTT) were obtained from AppliChem and Calbiochem, respectively. 1,1-diphenyl-2-picrylhydrazyl (DPPH), potassium persulfate, sodium carbonate and bromodeoxyuridine (BrdU) were purchased from Sigma-Aldrich (Steinheim, Germany). Mouse anti-BrdU antibody was acquired from Dako (Glostrup, Denmark). Vectashield mounting medium for fluorescence with 4′,6-diamidino-2-phenylindole (DAPI) was acquired from Vector Laboratories Inc., Peterborough, UK. Merck (Darmstadt, Germany) supplied dimethyl sulphoxide (DMSO), trichloroacetic acid (TCA) and Folin-Ciocalteu (F-C) reagent, whereas methanol was from Fisher Scientific (Loughborough, UK). FITC-conjugated Annexin V/propidium iodide (PI) assay kit was acquired from Cayman Chemical Company, USA. Silica gel (40–63 µm mesh; Merck, Kenilworth, NJ, USA) was used for column chromatographic separation, while silica gel 60 PF254 (Merck, Kenilworth, NJ, USA) was used for analytical (0.25 mm) TLC. DMSO-d6 (Aldrich, Steinheim, Germany) was used as solvent for 1H and 13C NMR spectra acquisition and TMS (Aldrich, Steinheim, Germany) was used as internal standard. 1D and 2D NMR spectra were recorded at Bruker Digital Avance 800 MHz spectrometer. Additional reagents and necessary solvents were purchased from VWR International (Leuven, Belgium).

Sampling

Samples of C. tamariscifolia, C. humilis and C. usneoides were collected in the middle/lower intertidal areas, during the low tide, between May and September 2012 on the Portuguese coast. Identification of specimens was made by Dr. Aschwin Engelen (Centre of Marine Sciences, University of Algarve, Portugal) and Dr. Javier Cremades Ugarte (Facultade de Ciencias, University of A Coruña). Voucher specimens of C. humilis (code number MB007), C. tamariscifolia (code number MB016) and C. usneoides (code number MB013) are deposited at the Centre of Marine Sciences, University of Algarve. Biomass was cleaned with distilled water (dH2O) and macroscopic epiphytes and extraneous matter were carefully removed. Samples were freeze-dried and stored at −20 °C until the extraction procedure.

Preparation of the extracts

The extracts were prepared by sequential extraction with solvents of increasing polarities index (PI), namely hexane (PI = 0.1), diethyl ether (PI = 2.8), ethyl acetate (PI = 4.4) and methanol (PI = 5.1). Biomass was mixed with hexane (1:10, w/v) and homogenized for 2 min using a disperser IKA T10B Ultra-Turrax at room temperature (RT). The tubes were then vortexed for 1 min, centrifuged (5,000 g, 10 min, RT) and the supernatants recovered. The extraction procedure was repeated 3 times and the supernatants combined and filtered. The residue was then sequentially extracted with diethyl ether, ethyl acetate and methanol as described above. The organic extracts were dried at 40 °C under vacuum. All extracts were dissolved in DMSO for biological activities screening or in the adequate solvent for chemical characterization, aliquoted and stored (−20 °C).

Total phenolic content (TPC)

TPC was determined using the F-C colorimetric method (Velioglu et al., 1998). Briefly, 5 µL of the extracts at the concentration of 10 mg/mL were mixed with 100 µL of a 10-fold diluted F–C reagent, incubated at RT for 5 min and mixed with 100 µL of sodium carbonate (75 g/L, w/v). After a 90 min incubation period at RT, absorbance was measured at 725 nm on a microplate reader (Biotek Synergy 4). The amount of TPC was calculated as gallic acid equivalents (GAE) using a calibration curve prepared with gallic acid standard solutions, and expressed as GAE in milligrams per gram of dried extract.

Antioxidant activity

Radical-scavenging activity (RSA) against DPPH

RSA against the DPPH radical was determined according to the method described by Brand-Williams, Cuvelier & Berset (1995) adapted to 96-well microplates (Moreno et al., 2006). Samples (22 µL) at concentrations ranging from 0.125 to 10 mg/mL were mixed with 200 µL of DPPH solution (120 µM) in methanol and incubated in darkness at RT for 30 min. The absorbance was measured at 515 nm (Biotek Synergy 4; Biotek, Winooski, VT, USA) and results expressed as antioxidant activity (%) relative to a control containing DMSO and as half maximal inhibitory concentration (IC50, mg/mL). Butylated hydroxytoluene (BHT, E320) was used as a positive control at the same concentrations of the extracts.

RSA against ABTS

RSA against ABTS was evaluated according to Re et al. (1999). A stock solution of ABTS∙+ (7.4 mM) was prepared in potassium persulfate (2.6 mM) as the oxidizing agent, and placed in darkness for 12–16 h at RT. The ABTS∙+ solution was diluted with ethanol down to an absorbance of 0.7 units at 734 nm on a Biotek Synergy 4 microplate reader. Samples (10 µL) at concentrations ranging from 0.125 to 10 mg/mL were mixed with 190 µL of ABTS∙+ solution in 96-well flat bottom microtitration plates, and 6 min upon mixing absorbance was read at 734 nm. Results were expressed as antioxidant activity (%) relative to a DMSO-containing control and as IC50 values (mg/mL). BHT was used as a positive control at the same concentrations of the extracts.

Cell lines and culture conditions

Human hepatocellular carcinoma HepG2 (ATCC® HB-8065™ ), human cervix adenocarcinoma HeLa (ATCC® CCL-2™ ), human gastric adenocarcinoma AGS (ATCC® CRL-1739™ ) and human colorectal adenocarcinoma HCT-15 (ATCC® CCL-225™) cell lines were maintained in RPMI-1640 culture media supplemented with 10% FBS (v/v), L-glutamine (2 mM), penicillin (50 U/mL) and streptomycin (50 µg/mL). Murine bone marrow stromal S17 cell line was kindly provided by D. Rawlings, UCLA, Los Angeles, CA. The latter cell line as well as human umbilical vein endothelial HUVEC (ATCC® CRL-1730™) and human neuroblastoma SH-SY5Y (ATCC® CRL-2266™ ) cell lines were grown in DMEM culture media supplemented with 10% FBS (v/v), L-glutamine (2 mM), penicillin (50 U/mL) and streptomycin (50 µg/mL). All cells were grown in an incubator at 37 °C and 5.0% CO2 in humidified atmosphere.

In vitro cytotoxic activity and selectivity

In vitro cytotoxic activity of the extracts was assessed by the MTT colorimetric assay (Mosmann, 1983). Briefly, exponentially growing cells were seeded at a density of 5 × 103 cells/well on 96-well plates and incubated for 24 h at 37 °C in 5.0% CO2. The extracts were then applied at concentrations ranging from 125 to 3.9 µg/mL for 72 h and cytotoxicity was evaluated. Positive and negative control cells were treated for 72 h with etoposide at the same concentrations of the extracts and DMSO at the highest concentration used in the test wells (0.5%, v/v), respectively. Two hours before the end of the incubation period, 20 µL of MTT (5 mg/mL in PBS, w/v) were added to each well and further incubated for 2 h at 37 °C. The optical density (OD) was measured on a Biotek Synergy 4 spectrophotometer at 590 nm. Results were expressed in terms of cell viability (%) and as half maximal inhibitory concentration (IC50, µg/mL). The selectivity index (SI) of the extracts was determined using the equation SI =CT∕CNT, where CT and CNT correspond to the extract-induced cytotoxicity on tumor (e.g., HepG2) and non-tumor cells (e.g., S17), respectively (Oh et al., 2011).

Cellular proliferation analysis by the BrdU incorporation assay

The effect of the extracts on HepG2 cells proliferation was evaluated by the BrdU incorporation assay. HepG2 cells were incubated for 72 h with complete medium, DMSO (0.5%, v∕ v), or with CTH at the concentrations of 2.31 or 4.62 µg/mL, which were the IC50 or 2 × IC50 concentration previously determined by the MTT assay. After a 1 h pulse with 10 µM BrdU, cells were washed with phosphate buffer saline (PBS), fixed in 4% paraformaldehyde in PBS, and cytospins prepared. After incubation in 2M HCl for 20 min, cells were incubated with mouse anti-BrdU (1:10, v/v) and further incubated with fluorescein-labeled rabbit anti-mouse antibody (1:100, v/v). For nuclear staining, Vectashield mounting medium for fluorescence with DAPI was used. Cells were observed in a LEICA DM2000 microscope using a 200 × magnification, and a semi-quantitative evaluation was performed by counting a minimum of 500 cells per slide.

Detection of apoptosis

Flow cytometry apoptosis detection through Annexin V-FITC staining

Apoptotic cells were identified and quantified by flow cytometry using the FITC-conjugated Annexin V/PI assay kit, according to the manufacturer’s instructions. Briefly, cells were treated for 72 h with complete medium, DMSO (0.5%, v∕ v), or with CTH at the concentrations of 3.9, 7.8 and 15.6 µg/mL. Etoposide treated-cells at IC50 concentration (1.85 µg/mL) were used as positive control. HepG2 cells were washed with ice-cold PBS, resuspended in 100 µL binding buffer, and stained with 5 µL of FITC-conjugated Annexin V (10 mg/mL) and 10 µL of propidium iodide PI (50 mg/mL). The cells were incubated for 15 min at RT in the dark and then 500 µL of binding buffer was added. Flow cytometry was performed using a FACS Calibur Flow Cytometer (Becton-Dickinson, East Rutherford, NJ, USA) and data acquisition and analysis were done with CellQuest Pro software. At least 1 × 104 events were recorded for each sample and represented as dot plots. For analysis, HepG2 cells were gated separately according to their size and granularity on forward scatter vs. side scatter plots. Apoptosis was evaluated on fluorescence channel 2 (for PI) vs. fluorescence channel 1 (for Annexin) plots (Zhang et al., 1997; Abu Bakar et al., 2010).

DAPI staining

HepG2 cells were grown in 6-well plates at seeding densities of 5 × 105 cells/well and treated for 72 h with CTH at 3.9, 7.8 and 15.6 µg/mL. Cells incubated with culture medium or with DMSO at the concentration of 0.5% (v/v) were used as blank or negative control, respectively. Etoposide treated-cells at IC50 concentration (1.85 µg/mL) were used as positive control. Cells were then washed with PBS and incubated with DAPI (5 µg/mL in PBS) for 2 min at RT. Fluorescence was visualized using a Leica DM LB (Leica Microsystems DI, Cambridge, UK) microscope, magnification 400×. Images were acquired using a Leica DC 300 FX digital camera. Cells under apoptosis were identified by marked condensation of chromatin and cytoplasm (apoptotic cells), plasma membrane blebbing (apoptotic bodies), and intra- and extracellular chromatin fragments (Murugan et al., 2010).

Determination of cytotoxic activity in a 3D multicellular tumor spheroids model (MCTS)

Generation of MCTS

HepG2 cells were used to produce spheroids by modification of the hanging drop method (Keller, 1995). Single-cell suspensions (1 × 104 cells/mL) were generated from trypsinized monolayers. Aggregate culture of HepG2 cells were generated by growth on non-adherent, bacterial-grade polystyrene Petri dishes. Cell suspension (30 mL) was then dispensed into 6 drops into the lid of a Petri dish. Upon inversion of the lid, the hanging drops were held in place by surface tension and cells accumulated at the free liquid–air interface. The Petri lids were placed in the dishes with PBS and incubated for four days under standard conditions.

MCTS treatment with bioactive extract

After four days, in each Petri dish, three of the six multicellular tumor spheroids (MCTS) were incubated with CTH at 20, 40 and 80 µg/mL for 24 and 48 h. Incubation was carried out by replacing the medium with 30 µL of fresh culture medium containing the extract. The remaining three MCTS were used as control; the cultured medium was replaced by fresh medium containing the same volume of DMSO. Images were captured at incubation time 0, 24 and 48 h by means of an Olympus SZX7 microscope (using a 20× magnification) with a digital camera (Optica B3). Each experiment was done in triplicate.

Compound isolation and elucidation

CTH (9 g) was fractionated by column chromatography (2.5 × 18 cm) over silica gel (SiO2) using increasing amounts of ethyl acetate in hexane (9:1; 85:15; 4:1; 75:25; 7:3; 3:2; 1:1) and increasing amounts of methanol in ethyl acetate (9:1; 8:1; 5:1; 2:1; 1:1), methanol (100%) and water (100%) as eluents to give 57 fractions. Each fraction was analyzed by TLC and pooled together to afford 21 samples. These samples were tested for cytotoxic activity and selectivity and the active fraction 7 (21.6 mg) was chosen for characterization. Fraction 7 was re-fractionated over SiO2 eluted with hexane; hexane/EtOAc (8:2); hexane/EtOAc (7:3); hexane/EtOAc (6:4); hexane/EtOAc (5:5); hexane/EtOAc (4:6); EtOAc and MeOH to afford compound 1 (1.1 mg).

Compound 1. Oil. 1H NMR (DMSO-d6, 500 MHz) δ 8.50 (1H, s, 4′-O H), 6.34 1H, br s, H-5′), 6.25 (1H, d, J = 3.0 Hz, H-3′), 6.20 (1H, d, J = 5.0 Hz, H-14), 5.57 (1H, d, J = 5.0 Hz, H-13), 4.29 (1H, s, H-6), 2.71 (2H, t, J = 7.5 Hz, H-1), 2.16 (2H, s, H-4), 2.07 (3H, s, 6′-C H 3), 1.90–1.20 (6H, m, H-8–H-10), 1.79 (2H, m H-2), 1.31/1.28 (3H, s, H-20), 1.25 (3H, s, H-17), 1.24 (3H, s, H-16), 1.23 (3H, s, H-19), 0.83 (3H, s, H-18); 13C NMR (DMSO-d6, 125 MHz) δ 149.4 (C-4′ ), 146.3 (C-1′), 143.9 (C-5), 140.1 (C-13), 126.2 (C-6′), 125.7 (C-14), 120.7 (C-2′), 115.5 (C-12), 114.6 (C-3′), 112.4 (C-5′), 110.4 (C-6), 87.8 (C-15), 74.8 (C-3), 45.6 (C-4), 43.6 (C-11), 43.2 (C-7), 42.5 (C-8), 35.6 (C-10), 30.8 (C-2), 28.5 (C-17), 26.3 (C-18), 24.9 (C-20), 22.6 (C-1), 21.9 (C-16), 19.9 (C-9), 19.6 (C-19), 15.9 (6′- C H3); LRESIMS m∕z 425 [M+ H]+. Chemical shifts are reported in δ units (parts per million) and coupling constants (J) in Hertz.

Statistical analysis

Results were expressed as mean ± standard error of the mean (SEM). Analysis of variance (ANOVA) was assessed using the SPSS statistical package for Windows (release 15.0, SPSS Inc.), and significance between means was analysed by the Tukey HSD test (p < 0.05). The IC50 values were calculated by sigmoidal fitting of the data by means of GraphPad Prism v. 5.0 (GraphPad Software, Inc., La Jolla, CA, USA). Pearson correlation coefficient (r) was also calculated (p < 0.01) to assess the strength of the linear relationship between two variables.

Results and Discussion

TPC and antioxidant activity

The results of total phenolic content and antioxidant activity are summarized in Table 1. C. tamariscifolia was the species with the highest TPC, mainly in the hexane, diethyl ether and ethyl acetate extracts, which presented TPC values higher than 100 mg GAE/g DW. C. usneoides diethyl ether extract also had a high TPC (122 mg GAE/g DW), whereas C. humilis methanol extract had the lowest levels of phenolic content (4.78 mg GAE/g DW). The highest RSA were observed with C. tamariscifolia ethyl acetate, diethyl ether and hexane extracts which IC50 for DPPH (IC50-DPPH) and for ABTS (IC50-ABTS) ranged from 0.17 to 0.63 mg/mL and from 0.26 to 0.52 mg/mL, respectively. Similar results were obtained with C. usneoides diethyl ether extract (IC50-DPPH = 0.65 mg/mL; IC50-ABTS = 0.60 mg/mL). The hexane and methanol extracts of C. humilis had the lowest scavenging activity (IC50 > 10 mg/mL for both radicals).

Table 1 Total phenolic content (TPC, mg GAE/g DW), and radical scavenging activity (RSA) on DDPH and ABTS radicals (IC50 mg/mL) of organic extracts of different species of Cystoseira.

Species/ compound	Extract	TPC (mg GAE/g DW)	IC50-DPPH (mg/mL)	IC50-ABTS (mg/mL)	
C. humilis	Hexane	24.42 ± 0.46e	>10	>10	
	Diethyl ether	20.34 ± 0.68e	8.28 ± 0.13d	8.85 ± 0.23d	
	Ethyl acetate	32.06 ± 0.72d	5.04 ± 0.13c	9.25 ± 0.43d	
	Methanol	4.78 ± 0.80f	>10	>10	
C. tamariscifolia	Hexane	113.13 ± 2.31b	0.63 ± 0.01a	0.52 ± 0.02a	
	Diethyl ether	116.61 ± 2.44b	0.30 ± 0.00a	0.47 ± 0.02a	
	Ethyl acetate	165.28 ± 1.92a	0.17 ± 0.00a	0.25 ± 0.01a	
	Methanol	45.04 ± 2.28d	1.08 ± 0.06b	2.93 ± 0.67b	
C. usneoides	Hexane	75.56 ± 0.21c	4.37 ± 0.03c	5.54 ± 0.06c	
	Diethyl ether	122.30 ± 0.81b	0.65 ± 0.01a	0.60 ± 0.01a	
	Ethyl acetate	17.76 ± 0.78e	7.37 ± 0.76d	>10	
	Methanol	17.03 ± 0.70e	7.16 ± 0.01d	>10	
BHT*		n.a.	0.07 ± 0.01	0.11 ± 0.00	
Notes.

Results are expressed as mean ± SEM of data obtained from six independent experiments.

a–f Different letters in the same column indicate significant differences by Duncan’s New Multiple Range Test at p < 0.05.

* positive control, 1 mg/mL.

n.a not applicable.

Taken together, our results indicate that C. tamariscifolia contains phenolic compounds of different polarities, which occur mainly in the less polar extracts. Distribution of phenolic compounds through different solvents may vary greatly usually due to their amphipathic properties and wide range of structures (Ivanova et al., 2005; Demiray, Pintado & Castro, 2009). Though commonly found in polar extracts such as methanol and water, phenolic compounds can also be present in less polar extracts including hexane, diethyl ether and ethyl acetate (Li et al., 2007; Maimoona et al., 2011). This may in fact explain the high levels of phenolic compounds in less polar extracts of Cystoseira, since the sequential extraction procedure used began with solvents of lower polarity (Li et al., 2007).

Phenolic compounds are described as strong antioxidants (Dai & Mumper, 2010). In this work, a significant correlation was observed between TPC and RSA on DPPH (r2 = 0.868, p < 0.01) as well as TPC and RSA on ABTS (r2 = 0.921, p < 0.01), suggesting that the antioxidant activity observed might be due to the activity of phenolic compounds. Data on TPC and antioxidant activity in macroalgae are scarce, but the Cystoseira genus generally has one of the highest total phenolic levels and antioxidant activities among Phaeophyceae macroalgae, such as Fucus serratus, Dictyota dichotoma, Bifurcaria bifurcata, Sargassum horneri and Alaria crassifolia among others (Zubia et al., 2009; Airanthi, Hosokawa & Miyashita, 2011). A few authors were able to relate the elevated antioxidant activity with tocopherol-like compounds, such as tetraprenyltoluquinol derivatives (Foti et al., 1994; Fisch et al., 2003). In addition, the high RSA obtained for Cystoseira extracts suggests that these macroalgae are potential sources of novel antioxidants that may help prevent oxidative stress and also an alternative to BHT and butyl-4-hydroxyanisole (BHA), two synthetic antioxidants found to be toxic and carcinogenic in animal models (Ito et al., 1986; Safer & Al-Nughamish, 1999).

Oxidative stress is considered to be one of the underlying causes of several chronic diseases, including cancer, and is implicated in both cytotoxic and apoptotic mechanisms (Goswami & Singh, 2006). The link between oxidative stress and cell death has been associated, for example, with lipid peroxidation, a process of oxidative degradation of lipids in which free radicals ‘remove’ electrons from membrane lipids. These events damage lipid bilayers, and impair several intra- and extra mitochondrial membrane transport systems, thus contributing to apoptosis. As a result, antioxidant compounds from natural sources have attracted much attention due to their ability to diminish oxidative stress. In fact, antioxidant compounds play an important role in regulation of gene expression and protection of DNA, lipids and proteins from oxidative stress-induced injury (Saura-Calixto, 2011). Because of this protective effect, it has been proposed that antioxidants may inhibit apoptosis when cancer cells should undergo cell death (Zeisel, 2004). However, the opposite has also been shown, i.e. molecules with known antioxidant properties have been described to also promote apoptosis (Moustapha et al., 2015). Therefore, the chemical structure of the antioxidant and its biological properties seemed to be essential to define the outcome of a given therapy with compounds with antioxidant properties.

Cytotoxic activity and selectivity

Natural extracts are considered as promising sources of antitumoral compounds when they exhibit IC50 values lower than 30 µg/mL (Dos Santos et al., 2010). This was the case for the hexane (CTH) and diethyl ether extracts of C. tamariscifolia, with IC50 values of 2.31 and 6.83 µg/mL, respectively (Table 2). In fact, in the literature, C. tamariscifolia also stood out as a potential source of antiproliferative compounds among other Phaeophyceae species (Zubia et al., 2009).

Interestingly, CTH had an IC50 statistically similar to that of the pure chemotherapeutic drug etoposide (IC50 = 1.85 µg/mL). This result indicates that CTH has cell growth inhibitory activity in vitro comparable with etoposide, a potent anti-cancer compound that acts as a topoisomerase II inhibitor (Scott & William, 2000). In fact, etoposide is one of the most potent drugs used in the treatment of several types of tumors, including testicular and ovarian cancer (Hande, 1998). However, different success rates are described for the treatment of different types of cancer with that compound. For example, Miao et al. (2003) reported the occurrence of resistant cell lines to this compound. It is also noteworthy to mention that HepG2 cells are known to display greater resistance to drugs and toxins comparing to other cells lines (Liu et al., 2010).

Table 2 In vitro cytotoxic activity, expressed as IC50 values (µg/mL) of organic extracts of different species of Cystoseira and etoposide on a human hepatocarcinoma cell line (HepG2).

Extracts	C. humilis	C. tamariscifolia	C. usneoides	Etoposide	
Hexane	>125	2.31 ± 0.08a	31.4 ± 3.22b	1.85 ± 0.12a	
Diethyl ether	>125	6.83 ± 0.01a	52.0 ± 3.19b	
Ethyl acetate	>125	44.2 ± 1.41b	>125	
Methanol	>125	>125	>125	
Notes.

Results are expressed as mean ± SEM of data obtained from six independent experiments.

a, b Different letters indicate significant differences by Duncan’s New Multiple Range Test at p < 0.05.

Since CTH had the highest cytotoxic activity towards HepG2 cells, this extract was further evaluated in other human tumor cell lines, namely cervical (HeLa), neuroblastoma (SH-SY5Y), gastric (AGS) and colorectal (HCT-15) carcinoma cells. Furthermore, CHT treatment was also carried out in murine stromal S17 and human umbilical HUVEC cell lines, both non-tumor cell lines, to determine the selectivity index (SI). As shown in Fig. 1, CTH had a strong cytotoxic activity in all tumor cell lines tested, except HeLa cells. This effect was, however, more pronounced towards HepG2 cells (IC50 = 2.31 µg/mL, p < 0.01 vs. S17 and HUVEC cells). Samples with SI values higher than 3 are deemed as highly selective (Mahavorasirikul et al., 2010). CTH was therefore considered highly selective when comparing HepG2 and S17 cells (SI = 5.5, Fig. 1) and especially against HUVEC cells (SI = 12.6). Based on these results, CTH was further used to study the mode of action associated with the cytotoxicity observed on HepG2 cells.

Figure 1 Effect of CTH on the viability of different cell lines.

(A) IC50 values of CTH on tumor cells (bars). Selectivity (scatter lines) was calculated using IC50 values of the non-tumor cell line S17 (∙) or HUVEC (∘) vs. the tumor cell lines. (B) IC50 value of CTH on non-tumor cell lines. Results are expressed as mean ± SEM of data obtained from six independent experiments, ∗p < 0.05, ∗∗p < 0.01 vs. HUVEC cells.

Cytotoxicity mechanisms

Cellular proliferation analysis by the BrdU incorporation assay

The BrdU incorporation assay was used in order to assess the effect of CTH on HepG2 cells proliferation. Results showed that cells treated with CTH at concentrations of 2.31 or 4.62 µg/mL incorporated less BrdU than control cells (treated with medium only or with 0.5% DMSO). In fact, the proliferation levels, expressed as the percentage of proliferating cells, underwent an almost 2-fold reduction, decreasing from 25.8% to 17.5% or 13.6% respectively after treatment with CHT at the concentration of 2.31 µg/mL or 4.62 µg/mL for 72 h (p < 0.01, Fig. 2).

Figure 2 BrdU incorporation (stained in green) with nuclei labelled with DAPI (stained in blue).

HepG2 cells (200 × magnification) were treated for 72 h with complete medium alone (A), 0.5% DMSO (B), or CTH at concentrations of 2.31 (C) or 4.62 µg/ml (D). Semi-quantitative analysis of BrdU incorporation was carried out by counting a minimum of 500 cells per treatment in each independent experiment (E). Results are expressed as the mean ± SEM of three independent experiments, ∗p < 0.05 vs. DMSO 0.5%. Scale bar = 100 µm.

Although previous data on the inhibition of cell proliferation with macroalgae extracts is very limited, it is interesting to observe that these results are consistent with studies from Funahashi et al. (1999). According to those authors, rats fed with commercial feed supplemented with wakame, an edible brown macroalga (Undaria pinnatifida) also belonging to the Phaeophyceae, showed significantly lower BrdU indices in tumor mammary cells as compared to a control group eating commercial feed alone. In fact, the authors showed that this phaeophyta had a strong suppressive effect on rat mammary carcinogenesis without toxicity, possibly via apoptosis induction.

Apoptosis-inducing activity

In order to verify whether CTH had apoptotic-inducing effect on HepG2 cells, two methodologies were applied: (i) analysis of the externalization of phosphatidylserine using flow cytometry (FITC-conjugated Annexin V/PI assay) and (ii) visualization of morphological alterations following DAPI staining.

Figure 3 Incubation of HepG2 cells with CTH promotes apoptosis.

Hepatocytes were treated with medium alone as blank (A), DMSO 0.5% (B, control), or CTH at concentrations of 3.9 (C), 7.8 (D) or 15.6 µg/mL (E) for 72 h. Hepatocytes were then stained with PI/Annexin V-FITC and analyzed by flow cytometry. (F) Quantitative analysis of apoptotic cells. Solid bars and errors represent the mean ± SEM, respectively (n = 6), ∗p < 0.01 vs. DMSO 0.5%.

During apoptosis there is a loss of membrane asymmetry due to the translocation of phosphatidylserine from the inner to the outer layer of the cell membrane (Koopman et al., 1994). This translocation occurs before nuclear breakdown and DNA fragmentation (Koopman et al., 1994; Wu, Ng & Lin, 2005). Since Annexin V strongly binds to phosphatidylserine, Annexin V binding to cells is considered to be a major marker of apoptosis (Zhang et al., 1997). The FITC-conjugated Annexin V/PI assay is a well-established method for the detection of living cells in early and late apoptosis. The four different quadrants of flow cytometric data represent four different states of cells. The lower left (LL) quadrant shows Annexin-/PI-normal healthy cells. The lower right (LR) and upper right (UR) represent early (Annexin +/PI-) and late apoptotic (Annexin+/PI+) cells, respectively. On the upper left quadrant (UL), necrotic (Annexin-/PI+) cells are displayed.

In this study, treatment of HepG2 cells with CTH resulted in a 2, 4 and 5-fold increase in the number of apoptotic cells, at the concentrations of 3.9, 7.8 and 15.6 µg/mL respectively (Fig. 3). Necrotic cells were also observed, but mostly after incubation with the highest concentration tested (15.6 µg/mL; 7.12 %). In fact, it has been described that treatment with cytotoxic drugs might stimulate apoptosis at lower doses and necrosis at higher doses (Zong & Thompson, 2006). Etoposide treated-cells (positive control) demonstrated 36.07 % of apoptotic cells after 72 h. These results indicate that apoptosis contributed significantly to the reduction in HepG2 viability when exposed to CTH. Moreover, the morphological alterations observed upon DAPI staining confirmed the results of the FITC-conjugated Annexin V/PI assay. In fact, treated cells exhibited noticeable morphological alterations typical of apoptosis, such as nuclear fragmentation and chromatin condensation (Fig. 4). These morphological modifications were dose-dependent and already visible after treatment with the lowest concentration tested (3.9 µg/mL).

Figure 4 HepG2 cells exposed to CTH showing apoptotic features.

Representative images (400× magnification) in which hepatocyte nuclei are stained with DAPI (in blue) are shown. Hepatocytes were treated with medium alone as blank (A), DMSO 0.5% (B, control), or CTH at concentrations of 3.9 (C), 7.8 (D), 15.6 µg/mL (E) or 1.85 µg/mL etoposide (F) as a positive control for 72 h. Scale bar = 50 µm.

Failure of apoptosis is a characteristic of the tumorigenic process. Thus, one strategy underlying anticancer drug development is the induction of the apoptotic machinery in cancer cells. In fact, most cytotoxic compounds used for cancer treatment are apoptotic inducers (Vecchione & Croce, 2010). On that note, recent research has shown strong evidence for anti-proliferative, pro-apoptotic and growth-inhibiting properties of Phaeophyceae extracts in a number of tumor models, including melanoma, lymphoma and lung cancer (Aisa et al., 2005; Culioli et al., 2004; Dias et al., 2005). Taken together, these results strongly indicate that C. tamariscifolia contains compounds that are able to induce apoptosis in a human hepatocarcinoma cell line.

Effect on MCTS

Anticancer drugs must penetrate into tumor cell masses to reach all cells at adequate concentrations. According to the vast majority of literature reports, many treatments are expected to lose efficacy in a three-dimensional (3D) pathophysiological environment, and testing on in vitro spheroid tumors is often considered a useful tool for negative selection to reduce animal testing or to evaluate drug candidates with enhanced tissue distribution and efficacy (Hirschhaeuser et al., 2010).

The effect of the application of CTH on 3D MCTS was examined in detail by optical microscopy. As shown in Fig. 5, MCTS presented a homogeneous size distribution in the controls. Loss of spheroid integrity was observed after 24 and 48 h following application of the extract at a concentration of 20 µg/mL. After incubation with 40 µg/mL of extract, this outcome was more evident and incubation with the hexane extract at 80 µg/mL, total disaggregation of MCTS occurred.

Figure 5 MCTS aggregation and morphology was influenced by CTH (20× magnification).

Control MCTS, growing as a suspension of multicellular aggregates, are shown with no incubation. The multicellular aggregates dissociated with 20, 40 and 80 µg/ml incubations for 24 and 48 h. Scale bar = 200 µm.

Generally, tumor cell lines are more resistant to antineoplastic agents when the cells are grown as spheroids rather than as monolayer cultures. The resistance of MCTS to anticancer drugs appears to reflect both limited drug penetration into the inner regions of the 3D cell masses as well as acquired resistance at the multicellular level (Gong et al., 2015). Although in vivo, tumors are affected by other cell types such as fibroblasts monocultures of multicellular spheroids from human tumor cell lines have proven to be a prevailing tool in the study of the micro-environmental regulation of tumor cell physiology and therapeutic problems associated with metabolic and proliferative gradients in a 3D cellular context (Rodriguez-Enriquez et al., 2008). The fact that the whole mass of tumor cells completely lost their adherence demonstrates that the compounds present in CTH have penetrated and may be effective in a multicellular tumor stage. The observed results combined with the anti-proliferative data confirmed the potential of CTH as a promising source of anticancer compounds.

Compound isolation, structural elucidation and bioactivities

CTH was subjected to a bio-guided fractionation, affording 21 fractions, which were tested for cytotoxicity at 20 µg/mL against HepG2 using the MTT assay (Fig. 6). Fractions 7, 9, 13 and 14 were those that strongly reduced the viability of HepG2 cells (Fig. 6, p < 0.001). Among these, fraction 7 was the one combining a high effect on HepG2 cell viability and highest selectivity index comparing to S17 cells (SI = 5.6). Thus, fraction 7 was further purified in order to isolate and identify its major compound.

Figure 6 Effect of different fractions obtained from CTH, at a concentration of 20 µg/mL, on HepG2 and S17 cellular viability.

Results are expressed as % of viability relative to a control containing DMSO (0.5%, v/v). Solid bars and errors represent the average and SEM, respectively (n = 12). Selectivity (scatter lines) was calculated using IC50 values of the non-tumor cell line S17 vs. the tumor cell lines.

Compound 1 (Fig. 7) was obtained as an epimeric mixture at C-3. The 1H NMR spectrum (DMSO-d6) showed two coupled aromatic hydrogen atoms at δ 6.34 (br, s) and 6.25 (d, J = 3.0 Hz), assigned to H-5′ and H-3′, respectively. Two chemical shifts attributed to hydrogens linked to sp2 carbons were also observed at δ 4.29 (s, H-6), 5.57 (d, J = 5.0 Hz, H-13) and 6.20 (d, J = 5.0 Hz, H-14) while six methyl groups were detected at δ 2.07 (6′-C H 3), 1.31/1.28 (H-20), 1.25 (H-17), 1.24 (H-16), 1.23 (H-19), and 0.83 (H-18). The occurrence of a chromane moiety in the molecule was proposed due the cross peaks at d 2.58 (t, J = 7.5 Hz) and 1.79 (m), assigned to H-1 and H-2, respectively, as observed in the COSY spectrum. The 13C NMR data confirmed the structural similarity with a cystoketal derivative (Amico et al., 1984) mainly due to the signals attributed to carbons of aromatic ring C-1′–C-6′ (δ 112.4–149.4), to carbons of double bonds C-5 (δ 143.9), C-6 (δ 110.4), C-13 (δ 140.1), C-14 (δ 125.7) as well as to carbinolic carbons C-12 (δ 115.5), C-15 (δ 87.8), and C-3 (δ 75.8). LRESIMS spectrum showed the protonated molecular ion [M+ H]+ at m∕z 425, establishing the molecular formula C27H36O4. Data of the isolated compound was consistent with demethoxy cystoketal chromane (compound 1, Fig. 7), a meroditerpene previously isolated from C. amentacea (Valls et al., 1996), a species closely related to C. tamariscifolia.

Figure 7 Structure of compound 1(demethoxy cystoketal chromane).

Compound 1 was evaluated for antioxidant activity at the concentration of 1 mg/mL, and had an activity of 18.21% and 13.73% towards DPPH and ABTS radicals, respectively. These results indicate that compound 1 was not responsible for the antioxidant activity detected in the crude extract. In fact, the antioxidant activity is most likely a result of a synergistic effect between different constituents of the crude extract as described by Palafox-Carlos et al. (2012).

Finally, compound 1 was tested towards HepG2 and S17 cells and was able to significantly reduce the viability of HepG2 cells (IC50 = 14.77 µg/ml) while maintaining a high selectivity towards S17 (IC50 = 48.46 µg/ml, SI = 3.28). Meroditerpenoids consist of a polyprenyl chain attached to hydroquinone ring moiety. In the marine environment, these compounds are especially abundant in brown algae such as species belonging to the Cystoseira and Sargassum genera (Blunt et al., 2014). In addition, various diterpenes have been identified as bioactive in C. crinita (Fisch et al., 2003), C. myrica (Ayyad et al., 2003) and C. usneoides (Urones et al., 1992). Furthermore, brown algal-derived chromene metabolites are reported to have anticancer and antimutagenic activities (Ayyad et al., 2011).

To date, no bioactivities of this compound have been reported previously. To the authors’ knowledge, this is the first time that demethoxy cystoketal chromane has been isolated from C. tamariscifolia and described as antiproliferative in HepG2 cells. In the future this molecule could be structurally optimized in order to increase pharmacokinetic and pharmacodynamic parameters among others.

Conclusions

In this work, three Cystoseira species were evaluated for their potential as sources of antioxidant and cytotoxic compounds. C. tamariscifolia had a strong antioxidant potential and a high content of phenolic compounds as well as a potent selective cytotoxic effect against hepatocellular carcinoma cells, especially its hexane extract (CTH). Moreover, CTH reduced cell proliferation and inhibited cell growth through apoptosis induction. This extract also had promising results in a 3D MCTS model, promoting the disaggregation of the mass of tumor cells after 24 h. Using bioactivity-guided fractionation procedures, it was possible to isolate and identify demethoxy cystoketal chromane as the major compound of CTH, and its selective cytotoxicity towards the recalcitrant HepG2 cell line was confirmed. It is also important to mention that this is the first description of demethoxy cystoketal chromane (1) in C. tamariscifolia, which was fully characterized by analysis of MS and NMR spectral data. Overall, Cystoseira can be considered a valuable source of bioactive secondary metabolites and a promising source of health products.

The authors would like to thank Jesus Garcia (IRB—Institute for Research in Biomedicine) for the realization of the spectra.

Additional Information and Declarations

Competing Interests

Author Contributions

Data Availability

The authors declare there are no competing interests.

Catarina Vizetto-Duarte conceived and designed the experiments, performed the experiments, analyzed the data, wrote the paper, prepared figures and/or tables, reviewed drafts of the paper.

Luísa Custódio, João H.G. Lago, Luísa Barreira and João Varela conceived and designed the experiments, analyzed the data, contributed reagents/materials/analysis tools, wrote the paper, reviewed drafts of the paper.

Gerardo Acosta performed the experiments, analyzed the data, contributed reagents/materials/analysis tools, reviewed drafts of the paper.

Thiago R. Morais, Carolina Bruno de Sousa, Maria João Rodrigues and Raquel T. Lima performed the experiments, prepared figures and/or tables, reviewed drafts of the paper.

Katkam N. Gangadhar performed the experiments, analyzed the data, prepared figures and/or tables, reviewed drafts of the paper.

Hugo Pereira performed the experiments, wrote the paper, prepared figures and/or tables, reviewed drafts of the paper.

M. Helena Vasconcelos conceived and designed the experiments, analyzed the data, reviewed drafts of the paper.

Amélia P. Rauter and Fernando Albericio conceived and designed the experiments, analyzed the data, contributed reagents/materials/analysis tools, reviewed drafts of the paper.

The following information was supplied regarding data availability:

The biological voucher specimens that are deposited at the Centre of Marine Sciences, University of Algarve cannot be accessed through a URL; interested parties can request access by emailing the corresponding author.

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
