# Peer review of "Can macroalgae provide promising anti-tumoral compounds? A closer look at Cystoseira tamariscifolia as a source for antioxidant and anti-hepatocarcinoma compounds"

_PeerJ, doi:10.7717/peerj.1704_

## Round 0.1 · original submission · Major Revisions

Please, consider all the suggestions of the reviewers, in the revised manuscript.

Reviewer 1 ·

Basic reporting

This manuscript investigated antioxidant effect and anti-tumoral of the extracts and isolated compound. It contains some new data, but, in the present form is not acceptable for publication.

Experimental design

Major revision :
• Detection of apoptosis, Line 208, need to be specify what the compounds were used.
• Magnification should be mentioned in any relevant method as well as figure legend.
• No result of the positive control in the apoptosis induction study.
• The concentration used in both apoptosis induction studied were quite high. And % apoptosis from flow cytometry was relative low compared to DAPI staining. How would the author address this?

Validity of the findings

• Lack of discussion information about association of antioxidant, cytotoxic and apoptosis induction.
• The necrosis cell death should be discussed because relatively high dose and longer period of time were used in the study where cells were harshly subjected to be death.

Additional comments

Minor revision :
• Introduction section, Line 85, the authors have to use more relevant cited reference for the sentence "The anti-proliferative potential was screened on human hepatocellular carcinoma HepG2 cells, a cell line known to be recalcitrant to cytotoxic drugs (Liu et al., 2010)." Therefore, the authors need to check and cite proper references throughout the text.
• Some typo, format and correction throughout the manuscript such as reference section need to be correct. For example reference of Lamia Mhadhebi, line 594.
• Abbreviation can be used after the full name is provided at the first presentation.

Reviewer 2 ·

Basic reporting

The authors have made good attempt to isolate and identify the active compounds with promising anti-hepatocarcinoma activities from Cystoseira tamariscifolia. The manuscript is presented in an intelligible fashion and written in standard English. The manuscript contains new information and needs publication. However, certain modification is required before publication.

Experimental design

The experimental design is good and I have no comments on it.

Validity of the findings

Over all paper quality is good and may be published after minor modification.
1. Authors should provide the structure elucidation of compound 1 with BC or NOE correlation in supporting information.
2. In this study, authors explored the biological activity of crude extract (CTH). Authors claim that Cystoseira tamariscifolia is a source for antioxidant and antihepatocarcinoma compounds. However, author only isolated and identified the compound 1 from fraction 7 with antihepatocarcinoma activity. Compound 1 was tested towards HepG2 and S17 cells and was able to significantly reduce the viability of HepG2 cells (IC50 = 14.77 µg/ml) while maintaining a high selectivity towards S17 (IC50 = 48.46 µg/ml, SI = 3.28). However, authors did not describe any antioxidant activity of compound 1. It is necessary for the authors to test the antioxidant activity of compound 1 in order to achieve the goal of this object.

---

## Round 0.2 · accepted · Accept

The manuscript has been improved and is now accepted.